# Genome-wide association study identifies common variants associated with breast cancer in South African Black women

Mahtaab Hayat [1,2,17] ✉, Wenlong C. Chen [1,3,4,17], Chantal Babb de Villiers [5], Sang Hyuck Lee[6,7], Charles Curtis[6,7], Rob Newton[8,9], Tim Waterboer [10], Freddy Sitas[11,12,13], Debbie Bradshaw[11], Mazvita Muchengeti [3,14,15], Elvira Singh[3,15], Cathryn M. Lewis [6,16], Michele Ramsay [1], Christopher G. Mathew [1,5,16,18] & Jean-Tristan Brandenburg [1,4,18] ✉

Genome-wide association studies (GWAS) have characterized the contribution of common variants to breast cancer (BC) risk in populations of European ancestry, however GWAS have not been reported in resident African populations. This GWAS included 2485 resident African BC cases and 1101 population matched controls. Two risk loci were identified, located between *UNC13C* and *RAB27A* on chromosome 15 (rs7181788, $p = 1.01 \times 10^{-08}$) and in *USP22* on chromosome 17 (rs899342, $p = 4.62 \times 10^{-08}$). Several genome-wide significant signals were also detected in hormone receptor subtype analysis. The novel loci did not replicate in BC GWAS data from populations of West Africa ancestry suggesting genetic heterogeneity in different African populations, but further validation of these findings is needed. A European ancestry derived polygenic risk model for BC explained only 0.79% of variance in our data. Larger studies in pan-African populations are needed to further define the genetic contribution to BC risk.

Breast cancer (BC) is the most common cancer in women worldwide, and the second most common cancer in South Africa. In 2020, the global incidence of BC was 2.26 million cases, with 129,415 cases in sub-Saharan Africa (SSA)[1]. Both genetic and environmental factors contribute to the risk of BC, and genetic risk factors may account for up to 30% of all BC cases[2]. These include both rare variants with large effect sizes and common variants identified by genome-wide association studies (GWAS). The first BC GWAS was published 16 years ago[3],

[1]Sydney Brenner Institute for Molecular Bioscience, Faculty of Health Sciences, University of the Witwatersrand, Johannesburg, South Africa. [2]School of Molecular and Cell Biology, University of the Witwatersrand, Johannesburg, South Africa. [3]National Cancer Registry, National Health Laboratory Service, Johannesburg, South Africa. [4]Strengthening Oncology Services Research Unit, Faculty of Health Sciences, University of the Witwatersrand, Johannesburg, South Africa. [5]Division of Human Genetics, National Health Laboratory Service and School of Pathology, Faculty of Health Sciences, University of the Witwatersrand, Johannesburg, South Africa. [6]Social, Genetic and Developmental Psychiatry Centre, Institute of Psychiatry, Psychology & Neuroscience, King's College London, London, UK. [7]National Institute for Health and Care Research Maudsley Biomedical Research Centre, South London and Maudsley NHS Foundation Trust, London, UK. [8]MRC/UVRI and LSHTM Uganda Research Unit, Entebbe, Uganda. [9]University of York, University of York, York, UK. [10]Infections and Cancer Epidemiology, German Cancer Research Center (DKFZ), Heidelberg, Germany. [11]Burden of Disease Research Unit, South African Medical Research Council, Cape Town, South Africa. [12]UNSW International Centre for Future Health Systems, Sydney, NSW, Australia. [13]School of Population Health, University of New South Wales, Sydney, NSW, Australia. [14]South African DSI-NRF Centre of Excellence in Epidemiological Modelling and Analysis (SACEMA), Stellenbosch University, Stellenbosch, South Africa. [15]School of Public Health, Faculty of Health Sciences, University of the Witwatersrand, Johannesburg, South Africa. [16]Department of Medical and Molecular Genetics, Faculty of Life Sciences and Medicine, King's College London, London, UK. [17]These authors contributed equally: Mahtaab Hayat, Wenlong C. Chen [18]These authors jointly supervised this work: Christopher G. Mathew, Jean-Tristan Brandenburg. ✉e-mail: mahtaab.hayat@wits.ac.za; jean-tristan.brandenburg@wits.ac.za

and this approach has been successful in identifying more than 200 loci associated at genome-wide significance with BC[4].

Most GWAS of BC have been performed in non-African populations, with almost 80% of all GWAS done in populations of European ancestry[5]. A large study of BC in Asian and European populations detected significant ancestral differences in the frequencies and association strengths of risk variants, and also identified 32 risk loci which showed differences in association between estrogen receptor (ER) positive and ER negative BC, indicating potentially important differences in the etiology of breast cancer subtypes[4].

There is a substantial emerging literature on the genetics of BC in African American (AA) populations[6–8], particularly from a collaborative study of three AA consortia for BC genetics which included the GWAS in Breast Cancer in the African Diaspora (ROOT), the African American Breast Cancer (AABC) and African American Breast Cancer Epidemiology and Risk (AMBER)[9–11]. In 2013 a study of 67 known BC loci discovered in non-African populations was investigated in an AA population. Only seven signals showed suggestive evidence of replication ($p < 0.05$) in this AA dataset[12]. Similarly, suggestive associations were reported in a study of candidate loci and a GWAS that included participants from the ROOT and AABC consortia[10,13]. However, in a meta-analysis by Huo et al. three variants were associated with BC in women of African ancestry at genome-wide significance[7]. Two single nucleotide polymorphisms (SNPs), rs13074711 upstream of *TNFSF10* and rs10069690 in *TERT*, were associated with ER negative BC. The third, rs12998806, was associated with the risk of ER positive BC. Ruiz-Narvaez et al. used admixture mapping that included participants from the AMBER consortium to identify two novel associations, rs112545418 in *ZFYVE28* and rs55850050 on chromosome 17, with ER positive BC[14]. Another study that included participants from the AMBER consortium tested 65 SNPs for association with BC, but did not find any significantly associated SNPs[15]. A meta-analysis of African ancestry cohorts and European ancestry cohorts from Breast Cancer Association Consortium (BCAC) found four loci associated with overall BC risk (1p13.3, 5q31.1, 15q24 and 15q26.3) and two with ER negative BC (1q41 and 7q11.23), with modest contributions from the African cohorts[16]. Recently a large GWAS of BC cases and controls of African ancestry predominantly from the Unites States identified 12 loci associated with breast cancer risk which included a low frequency missense variant in the *ARHGEF38* gene and a common variant associated with triple negative breast cancer (TNBC)[17]. The sample sizes in the African-American GWAS in these studies ranged from 3153 BC cases and 2831 controls to the most recent study which included 18,034 cases and 22,104 controls[17].

In contrast to GWAS in AA populations no GWAS have been carried out exclusively in resident SSA populations. A number of small candidate gene association studies investigated the contribution of common variants to BC in SSA. Six of these studies were reviewed by Hayat et al., and three further studies were published more recently[8,18–20]. The sample sizes in these studies ranged from 40 to 392 cases and 39–250 controls, and none reported strong evidence for association with BC. A recent study examined four *FGFR2* SNPs which are associated with BC in populations of European or African American ancestry in 1001 cases and 1006 controls from southern African Black women and did not find evidence of association with BC[21].

GWAS has also led to the development of polygenic risk scores (PRS) for the stratification by BC genetic risk. Risk prediction tools, such as BOADICEA, developed in a European setting using both clinical and genetic data has demonstrated to be effective in the management of BC risk[22]. BC PRS are primarily developed using European genetic data, and previous studies have demonstrated poor transferability of European PRS to non-European populations[23,24]. This reinforces the need for population diverse GWAS for BC in order to develop PRS that are more appropriate.

Genotyping and whole genome sequencing studies have revealed a very high degree of genetic diversity among the populations of the African continent, with principal component analysis showing clear separation of populations from West, East, Central and Southern Africa[25]. African-Americans originated from Africans forced into slavery and are descended mostly from ethnic groups that lived in West Africa, with admixture mostly of European ancestry[26]. It is therefore likely that genetic studies of breast cancer in African-Americans will capture only a subset of the contribution of the genetic contribution to breast cancer susceptibility on the African continent, and argues for broadening the diversity of genetic studies in Africa. In view of the paucity of genetic research into the etiology of BC in Africa[8], and the genetic diversity of African populations[25,27], we carried out a GWAS to identify common genetic variants that contribute to BC risk in a South African Black population. This included cases and controls from the Johannesburg Cancer Study (JCS) and ethnically matched controls from the Africa Wits-INDEPTH Partnership for Genomic Research (AWI-Gen) study[27–30]. The JCS samples formed part of a larger study, Evolving Risk Factors for Cancer in African Populations (ERICA-SA) (https://www.samrc.ac.za/intramural-research-units/evolving-risk-factors-cancers-african-populations-erica-sa) which is investigating the contributions of lifestyle, infection and genetics to cancer. We also performed a meta-analysis of the African ancestry GWAS datasets from Jia et al.[17] and the UK Biobank (UKBB) to identify potential shared risk loci for populations of African ancestry. Finally, we examined the transferability of a BC PRS developed from populations of European ancestry to our dataset.

## Results

### Study participants, structure control and dataset

Although all participants were from the Soweto region of greater Johannesburg in South Africa, we controlled and adjusted for the population substructure that was present. Following population substructure analysis, 226 cases and 69 controls were removed, leaving 2485 cases and 1101 controls to be included in the association analysis (Table 1, Supplementary Dataset 1). PCs 1–5 accounted for most of the variance observed from the Eigenvalue curve (Supplementary Fig. S1) and were selected as covariates in the linear mixed model (LMM). The admixture plot (Fig. 1A) shows clear differences between West, East and South African populations. The PC plot showed that the South African BC cases and controls were well matched and clustered away from non-South African samples and that West African populations are distinct from South African populations (Fig. 1B). Finally, participant relatedness was accounted for with genetic relationship matrices (GRMs) that were generated with 500,000 markers using the leave-one-chromosome-out (LOCO) approach and used in the LMM (see Methods).

The total genotyping rate was 97.83% before data QC but improved to 99.92% after QC. The final dataset included 1,699,678 genotyped SNPs, and a total of 18,020,999 genotyped and imputed SNPs to be tested for association with BC.

### Genome-wide association analysis for BC in the South African population

SNPs were tested for association with BC using an LMM method which was used because it is effective in correcting for relatedness and structure, therefore limiting genetic inflation[31]. The genomic inflation factor (λ genomic control) for the model was 1.01 (Fig. 2A). Two signals that were significantly associated with BC in our dataset were identified. The first is a genotyped SNP on chromosome 15 that is located between the genes *UNC13C* and *RAB27A/RSL24D1* (rs7181788, $p = 1.01 \times 10^{-08}$). The second is an intronic variant within *USP22* (rs899342, $p = 4.62 \times 10^{-08}$) on chromosome 17 (Table 2, Fig. 2B). Regional association plots show that there are multiple correlated SNPs in the region of both signals (Fig. 3). The 95% credible set from

the FINEMAP analysis included the top signal on chromosome 15 and three SNPs on chromosome 17, particularly the top signal identified in this GWAS (Supplementary Dataset 2).

Additionally, 89 SNPs, from 39 independent loci were identified with suggestive association with BC ($p < 5 \times 10^{-06}$) (Supplementary

Dataset 3). The estimated genetic heritability (h2g) was 17.50% (standard deviation: 6.52%) on the liability scale.

### Replication of JCS associations in African ancestry BC GWAS

We first carried out a meta-analysis of African ancestry (AA) cases and controls from Jia et al.[17] and the UKBB (see Methods) to generate a joint AA data set (Supplementary Fig. S2). SNPs from the South African JCS BC GWAS with at least suggestive evidence of association ($p_{JCS} < 5 \times 10^{-6}$) were then assessed for replication in this joint dataset AA dataset. A subset of 33 independent markers from our JCS study were present in the AA dataset (Supplementary Dataset 4), none of which reached a Bonferroni p value threshold ($p < 1.52 \times 10^{-3}$), including the top hits from our study. Of the 33 markers, 20 had the same directional effect (exact binomial test $p = 0.296$).

### Replication of suggestive hits from AA BC GWAS in JCS GWAS

We then tested whether loci that were associated with BC ($p_{AA} < 5 \times 10^{-6}$) in the African Ancestry meta-analysis were associated with BC in the South African JCS GWAS. There were 54 independent loci in the AA meta-analysis, two of which met Bonferroni correction with same sign of the effect ($p_{JCS} < 9.3 \times 10^{-04}$) in the SA JCS data. These included 19 SNPs near *TOX3* on chromosome 16 led by rs3112570 ($p_{JCS} = 1.37 \times 10^{-04}$) and rs7734992 ($p_{JCS} = 3.44 \times 10^{-04}$) in *TERT* on chromosome 5, while several others had nominal evidence of association with

**Table 1 | Sample sizes**

| SA GWAS datasets | N | |
|---|---|---|
| BC GWAS cases | 2485 | |
| GWAS controls | 1101 | |
| ER-positive | 1155 | |
| ER-negative | 766 | |
| HER2-positive | 499 | |
| TNBC | 262 | |
| Other studies | Cases | Controls |
| UK Biobank (African Ancestry) | 163 | 3774 |
| Jia et al. (2024) (African American) | 18,034 | 22,104 |
| Total (Meta-analysis) | 18,197 | 25,878 |

*SA* South African, *ER-positive* estrogen receptor positive, *ER-negative* estrogen receptor negative, *HER2-positive* human epidermal growth factor positive, *TNBC* triple negative breast cancer.

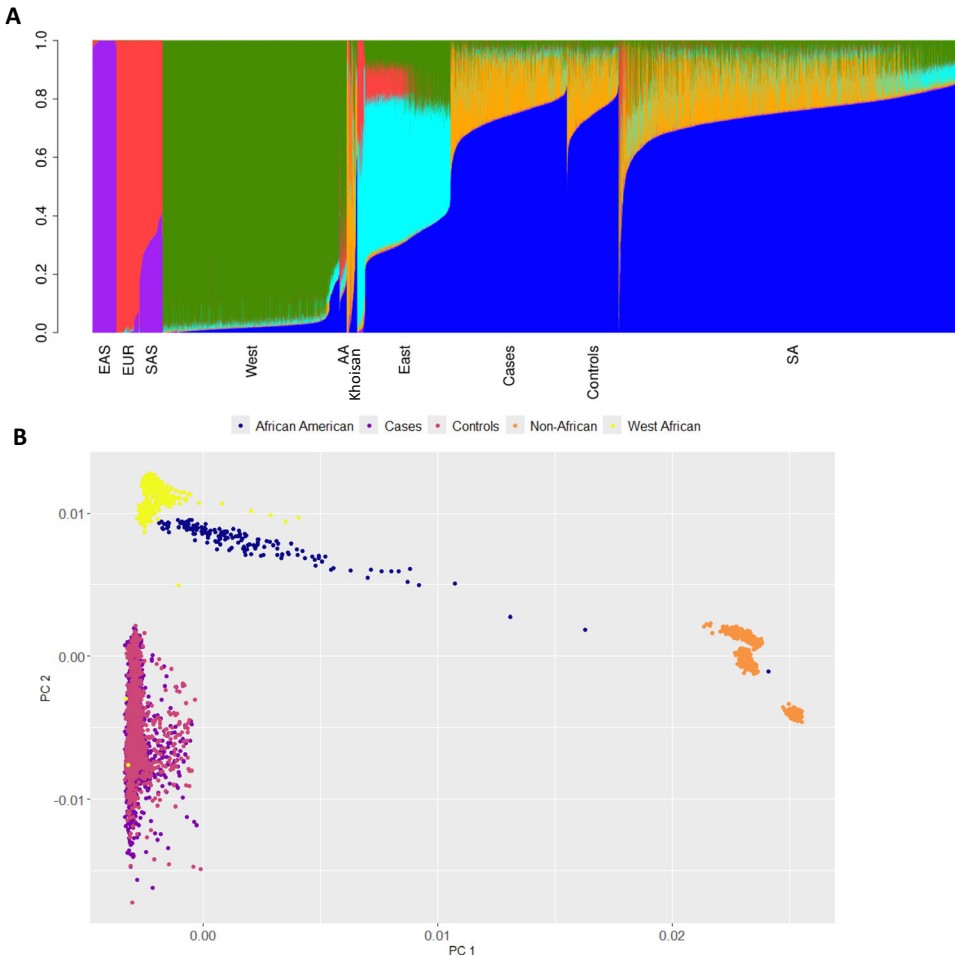

**Fig. 1 | Admixture and principal component plots. A** The Admixture plot at K = 6 of cases, controls and reference populations: East Asian (EAS) (KGP), South-East Asian (SAS) (KGP) and European (EUR) (KGP), West African (West) (KGP and AWI-Gen); African American (AA) (KGP); Khoe-San[55]; East African (East) (KGP, AGVP and AWI-Gen); South African (SA) (AGVP, AWI-Gen and JCS). **B** PCA Plot (1st and 2nd components) showing JCS cases and controls, African Americans, Non-African (CEU, SAS and EAS) and West Africans. KGP Thousand Genomes Project, AGVP African Genome Variation Project, JCS Johannesburg Cancer Study.

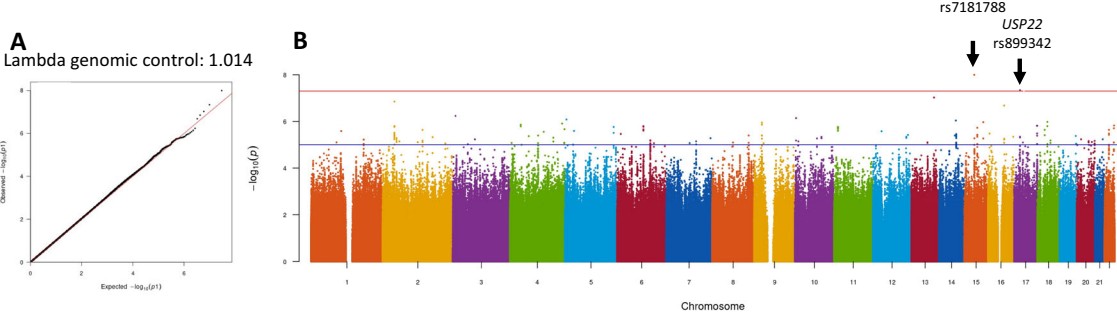

**Fig. 2 | Quantile-Quantile plot and Manhattan plot of the South African JCS BC association results. A** Quantile-Quantile (QQ) plot, λ = 1.014. **B** Manhattan plot with genome-wide significant hits ($p < 5 \times 10^{-08}$) highlighted on chromosomes 15 and 17. Red line indicates genome-wide significance ($p < 5 \times 10^{-08}$), blue line indicates suggestive significance ($p < 5 \times 10^{-06}$). JCS Johannesburg Cancer Study, BC breast cancer, λ Lambda genomic control.

$p < 0.05$ including loci on chromosomes 2, 3, 15 and 19 (Supplementary Dataset 5).

### Receptor sub-type analysis

Potential differences in the genetic etiology of breast cancer subtypes were investigated by additional association analyses of TNBC vs controls, TNBC vs ER-positive BC, TNBC vs HER2-positive BC, ER-positive and ER-negative BC vs controls, and ER-positive vs ER-negative BC.

The TNBC vs controls analysis identified six SNPs at two loci that reached genome-wide significance (Table 3, Fig. 4) including rs111999709, EAF = 0.058, $p_{TNBCvsctrl} = 2.08 \times 10^{-08}$ on chromosome 3, and rs11598380 on chromosome 10, EAF = 0.015, $p_{TNBCvsctrl} = 2.97 \times 10^{-08}$. The TNBC vs ER-positive BC analysis revealed one significantly associated SNP, rs189230042, on chromosome 6, EAF = 0.025, $p_{TNBCvsER+} = 2.33 \times 10^{-8}$ (Table 3, Supplementary Fig. S3). No genome-wide significant signals were found in the TNBC vs HER2-positive analysis (Supplementary Fig. S4) or in the analysis of ER-positive BC vs controls (Supplementary Fig. S5). Analysis of ER-negative BC vs controls showed one genome-wide significant signal on chromosome 10: rs11593018, $p_{ERneg} = 4.92 \times 10^{-08}$ (Table 3, Supplementary Figs. S6 and S7). This was supported by two other SNPs in close proximity. The next strongest, but not genome-wide significant, signal was rs7181788 on chromosome 15, which was the top signal in the overall BC GWAS (Supplementary Fig. S7B). The ER positive vs ER negative analysis identified genome-wide significant signals at two loci, on chromosomes 3 and 1 (Table 3, Supplementary Figs. S8 and S9A). The strongest signal was rs112965634, $p_{ERpvn} = 2.22 \times 10^{-08}$ on chromosome 3. Two SNPs on chromosome 1 reached genome-wide significance: rs113934974, $p_{ERpvn} = 3.06 \times 10^{-08}$ and rs113425481, $p = 3.09 \times 10^{-08}$ (Supplementary Fig. S9B). The associations at both of these loci were supported by multiple other SNPs in these regions (Supplementary Fig. S9).

Potential replication of signals with $p < 5 \times 10^{-06}$ in the JCS ER-negative subtype vs control analysis was assessed in the Jia et al. African data using their ER-negative vs control analysis[17].

The genome wide significant association on chromosome 10 in the JCS ER-negative vs controls analysis was not replicated in the Jia et al. African data; interestingly, the EAF in the Jia et al. data (0.051) was substantially higher than in the South African data (0.016). None of the suggestive associations in the JCS data were replicated in the Jia et al. African data (Supplementary Dataset 6). There were no genome-wide significant signals in the JCS ER-positive vs controls analysis, and the Jia et al. study did not include an analysis of ER-positive vs ER-negative subtypes.

Replication of ER-negative vs controls and ER-positive vs controls signals ($p < 5 \times 10^{-06}$) from Jia et al. were assessed in the JCS dataset. None of these met the Bonferroni threshold for either set of analyses ($p = 2.5 \times 10^{-04}$ and $p = 1.36 \times 10^{-04}$, respectively). However, consideration of only the three genome-wide significant loci from Jia et al. in the

ER-negative vs controls analysis found that SNPs at two of these loci, led by rs7734992 on chromosome 5 and rs11668840 on chromosome 19 showed evidence of association in the JCS data with $p_{JCS} = 7.55 \times 10^{-3}$ and $9.30 \times 10^{-04}$ respectively (Supplementary Dataset 7). Similarly, for the 7 genome-wide significant loci in the ER-positive vs controls in Jia et al., SNPs at two of these loci, on chromosome 2 (led by rs17778798, $p_{JCS} = 4.68 \times 10^{-04}$) and chromosome 16 (led by rs3112570, $p_{JCS} = 6.94 \times 10^{-04}$) also showed evidence of association (Supplementary Dataset 8).

### Functional analysis

The top signal on chromosome 15 at rs7181788 is flanked by potential candidate genes *UNC13C* and *RAB27A*. *RAB27A* is a member of the RAS oncogene family involved in exosome secretion and is associated with consequent invasive growth and metastasis. The top SNP on chromosome 17, rs899342, is located in an intron of *USP22*. This SNP is a strong eQTL for expression of *USP22* in a wide range of tissues, and on PancanQTL it affected the expression of *USP22* in lower grade gliomas and thyroid carcinoma. Data on GTEx shows that the C allele downregulates expression of *USP22* in the thyroid gland. Regarding the associations identified in the ER-negative subtype analysis, the nearest gene to the locus identified on chromosome 10 is *SGMS1* (sphingomyelin synthase 1), but no eQTL data is available for the associated SNPs at this locus. In the ER-positive vs ER-negative analysis, the eQTL analysis using FUMA showed that the SNPs rs113934974 and rs113425481 upstream of *TMEM52* on chromosome 1 are eQTLs for expression of this gene in mammary tissue ($p = 1.2 \times 10^{-6}$ and $8.1 \times 10^{-6}$ respectively).

Little is known of the function of *TMEM52*; it encodes a transmembrane protein and is positively regulated by p53 so may be involved in the cellular stress-response system[32].

### Polygenic risk score

A polygenic risk score was generated with 202 SNPs that were in our South African JCS GWAS dataset and in common with the 313 SNP PRS model from Mavaddat et al. (PRS313/202)[33]. This model explained only 0.79% of variance in our dataset, with an AUC of 0.56 (Fig. 5). A PRS was also generated with 2819 SNPs in common with the 3820 SNPs that Mavaddat et al. (PRS3820/2819) reported to have optimal predictability. This model explained only 0.6% of variance in our dataset with an AUC of 0.55 (Fig. 5).

### Discussion

Although a wealth of information now exists on the contribution of common genetic variants to susceptibility to breast cancer, the majority of genome-wide studies have been carried out in populations of European ancestry. There is also a burgeoning literature on the genetics of breast cancer in African American populations, but we are

**Table 2 | Top SNPs associated with African breast cancer from chromosome 15 and 17**

| rsID | Chr | Position (hg19) | Gene | Alleles (effect/non-effect) | EAF (cases) | EAF (controls) | OR (95% CI) | p value | Impute2 Score |
|---|---|---|---|---|---|---|---|---|---|
| rs7181788[a] | 15 | 55015367 | UNC13C, RAB27A | T/G | 0.252 | 0.19 | 1.41 (1.33–1.47) | $1.01 \times 10^{-08}$ | 1.00 |
| rs899342 | 17 | 20924620 | USP22 | T/C | 0.142 | 0.193 | 0.68 (0.67– 0.70) | $4.62 \times 10^{-08}$ | 0.98 |

SNP is considered genome-wide significant if p-value <5×10[-08].

EAF effect allele frequency, OR odds ratio, 95% CI 95% confidence interval SNP is considered genome-wide significant if p-value < 5×10[-08].

[a]Genotyped SNP OR and 95% CI calculated in reference to effect allele.

not aware of any genome-wide studies in resident African populations. The immense genetic diversity among the populations of sub-Saharan Africa and differences in environmental exposures between resident and non-resident African populations suggests that there may be substantial differences in the genetic determinants of cancer susceptibility both within continental Africa and across continents[34]. Bridging this knowledge gap is needed to increase our understanding of the genetic etiology of African breast cancer and to develop clinical tools such as polygenic risk scores that can guide screening approaches in Africa, and globally. Our genome-wide study in Black South African women is a step towards this goal.

Correcting for the complex genetic diversity and population substructure on a regional level was important in generating a robust dataset to be used in the association analysis. A substantial contribution of genetics to BC risk was observed in this population, with a SNP-based heritability (h2g) estimate of 17% in the South African JCS dataset. This is lower but comparable to the h2g estimate from an African Ancestry study of 22%[17].

The South African JCS GWAS identified two strongly associated genetic risk loci for BC in a South African Black population namely a risk allele rs7181788 on chromosome 15, which lies between the genes *UNC13C* and *RAB27A*, and a risk allele rs899342 within the *USP22* gene on chromosome 17. *RAB27A* is a small GTPase and member of the RAS oncogene family, with an important role in exocytosis. Overexpression of Rab27A protein has long been associated with increased invasive and metastatic abilities in breast cancer cells both in vitro and in vivo[35]. More recently, silencing of this gene was found to inhibit proliferation, invasion and adhesion of triple negative breast cancer cells[36]. Also, migration and invasion of colon cancer cells were shown to be suppressed by *RAB27A* knockdown but were promoted by *RAB27A* ectopic expression[37]. *UNC13C* is one of a family of proteins with key roles in exocytosis and has been reported to downregulate tumor progression in oral squamous cell carcinomas through its role in regulating epithelial-to-mesenchymal transition (EMT) signaling pathways[38]. A recent study found high numbers of mutations in *UNC13C* in head and neck cancer patients of African ancestry, which suggests these variations can lead to aggressive forms of head and neck cancer in patients of African ancestry[39].

The risk allele rs899342 lies within the *USP22* gene on chromosome 17 and affects expression of this gene in many tissues. *USP22* is a ubiquitin hydrolase and is a component of the SAGA coactivator complex which is essential for eukaryotic transcription. It is highly expressed in breast cancer samples compared to benign breast tissue, and high expression of *USP22* is significantly associated with poorer overall survival in breast cancer[40,41]. It also associates with estrogen receptor α to maintain ERα stability and contributes to chemotherapy resistance in triple negative BC tumors[40,41].

Receptor subtype analysis of TNBC vs controls revealed association with two loci in gene 'desert' regions, with the nearest genes being *IL2ORB* and *HACD1* on chromosomes 3 and 10 respectively. The TNBC vs ER-positive top signal on chromosome 6 is in a long non-coding RNA, with the nearest gene being *RGS17*, which is a negative prognostic marker for TNBC[42]. The ER-negative BC vs controls analysis revealed an intergenic signal that reached genome-wide significance and was supported by two other SNPs in close proximity. The nearest gene is *SGMS1*, a sphingomyelin synthase, which, if overexpressed in breast cancer cell lines, inhibits TGF-β1-induced EMT and the migration and invasion of cells[43]. Receptor subtype analysis for ER positive-BC did not detect any signals at genome-wide significance.

An analysis of ER-positive vs ER-negative BC cases was done to screen for genetic signals that are specific to a particular subtype. The signal from this analysis, rs112965634 on chromosome 3, is intergenic and is extremely rare in non-African populations. The nearest gene at this locus is the histone acetyltransferase *KAT2B*, which is upregulated by a transcriptional complex, NELF-E-SLUG, and promotes the EMT process in the development of breast cancer[44]. Inactivation of *KAT2B*

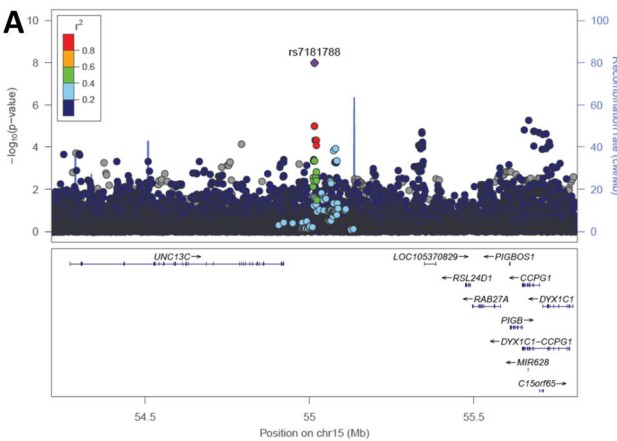
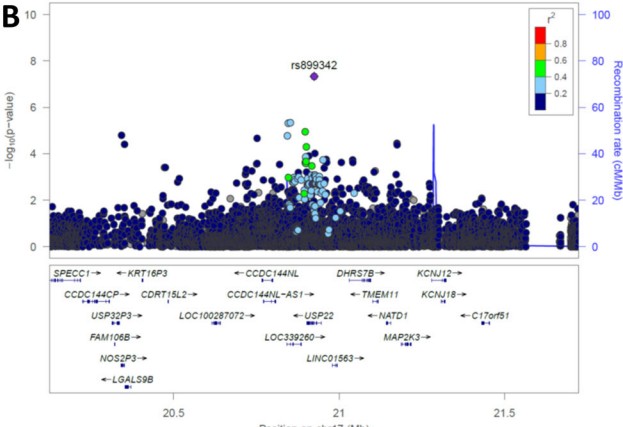

**Fig. 3 | Regional association plots of the top signals in the South African JCS GWAS using *locuszoom* software and the JCS Soweto as a reference for LD.** **A** rs7181788 on chromosome 15 between *UNC13C* and *RAB27A*. **B** rs899342 on chromosome 17 in *USP22*. JCS Johannesburg Cancer Study, LD linkage disequilibrium, GWAS genome wide association study.

## Table 3 | Top signals from receptor sub-type GWAS

| TNBC vs controls | | | | | | |
|---|---|---|---|---|---|---|
| Chr | Position (hg19) | rsID | Alleles (Effect/non-effect) | EAF | *P* value | OR (95% CI) |
| 3 | 137142030 | rs111999709 | C/T | 0.058 | $2.08 \times 10^{-08}$ | 1.20 (1.13–1.28) |
| 3 | 137126170 | rs534829894 | A/G | 0.058 | $2.74 \times 10^{-08}$ | 1.20 (1.13–1.28) |
| 3 | 137142198 | rs113378419 | T/C | 0.058 | $2.95 \times 10^{-08}$ | 1.20 (1.12–1.28) |
| 3 | 137124646 | rs111295639 | C/G | 0.063 | $3.79 \times 10^{-08}$ | 1.19 (1.12–1.27) |
| 3 | 137126730 | rs112262998 | A/G | 0.063 | $3.93 \times 10^{-08}$ | 1.19 (1.12–1.27) |
| 10 | 17669070 | rs11598380 | T/C | 0.015 | $2.97 \times 10^{-08}$ | 1.45 (1.27–1.66) |
| TNBC vs ER-positive | | | | | | |
| 6 | 153702044 | rs189230042 | A/T | 0.025 | $2.33 \times 10^{-08}$ | 1.36 (1.22–1.52) |
| ER-negative vs controls | | | | | | |
| 10 | 52055245 | rs11593018 | A/G | 0.016 | $4.92 \times 10^{-08}$ | 0.23 (−0.08–0.54) |
| 10 | 52054031 | rs7073005 | T/C | 0.016 | $7.76 \times 10^{-08}$ | 0.23 (−0.09–0.54) |
| 15 | 55015367 | rs7181788 | T/G | 0.219 | $3.60 \times 10^{-07}$ | 1.49 (1.32–1.72) |
| ER-positive vs ER-negative | | | | | | |
| 3 | 20660927 | rs112965634 | G/C | 0.060 | $2.22 \times 10^{-08}$ | 0.46 (0.35–0.57) |
| 1 | 1851188 | rs113934974 | G/A | 0.503 | $3.06 \times 10^{-08}$ | 1.44 (1.39–1.49) |
| 1 | 1851185 | rs113425481 | T/G | 0.503 | $3.09 \times 10^{-08}$ | 1.44 (1.39–1.49) |

SNP is considered significant if *p* value < $5 \times 10^{-08}$.
*EAF* effect allele frequency, *OR* odds ratio, *95% CI* 95% confidence interval.

was associated with downregulation of the EMT pathway, whereas elevated expression of *KAT2B* was correlated with reduced survival in breast cancer patients. The SNPs in the locus identified on chromosome 1 in the ER-positive/ER-negative analysis are located just upstream of the *TMEM52* gene and are eQTLs for its expression in mammary tissue, but this locus has not previously been reported to be associated with BC.

There was limited evidence for replication of our GWAS findings in the African Ancestry datasets. Our top signals from the overall GWAS on chromosome 15 and chromosome 17 were not replicated in the AA data meta-analysis. The lack of shared risk loci could be explained in part by African Americans mostly being descended from West African populations with European admixture while the South African JCS GWAS was composed of cases and controls from South Africa, who are South-Eastern Bantu-speaking populations with Koisan admixture[27]. Our admixture analysis and PCA plot shows very substantial genetic diversity between West African and South African populations. Also there are potential differences in environmental exposures between these populations. However, the lack of replication in the AA dataset requires further investigation as they may be false positives. Some, but

not all the signals from the African ancestry meta-analysis dataset were replicated in our JCS dataset, which could be attributed to both genetic diversity and the limited power for replication in our dataset.

The PRS models evaluated in our study showed that models generated in European populations had substantially lower predictive efficacy for BC in the South African JCS population, with AUCs of 0.56 and 0.55 for the $PRS_{313/202}$ and $PRS_{3820/2819}$ respectively as compared to 0.63 and 0.64 in the European ancestry study[33]. The 313 SNP PRS also did not perform well on the Jia et al. African Ancestry dataset with an AUC of 0.58[17]. This is consistent with findings on the performance of PRS findings in other disorders[5,23,24]. More GWAS need to be carried out in resident African populations to generate more predictive PRS, the inclusion of diverse populations in PRS generation can improve the transferability of risk loci and PRS across different populations[45]. PRSs have been shown to have attenuated risk prediction both in discrimination and calibration when used in non-European ancestry populations. PRS represents a significant advance in BC risk prediction, with potential for further enhancing personalized care[46]. The role of PRS in the clinical management of BC is being extensively researched,

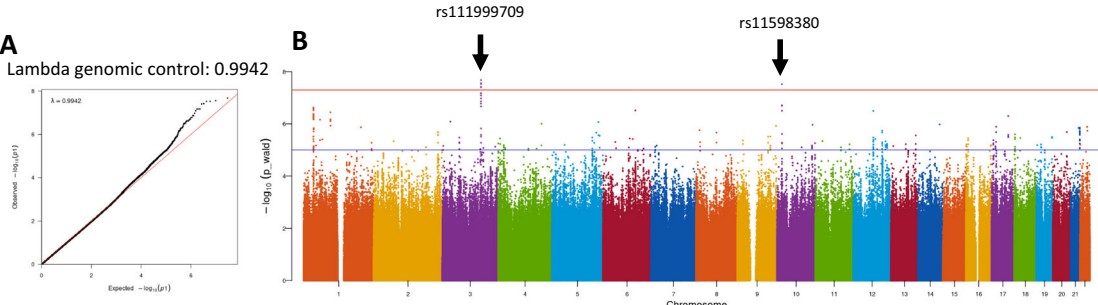

**Fig. 4 | Quantile-Quantile plot and Manhattan plot of the TNBC vs controls analysis. A** QQ plot, λ = 0.9942. **B** The Manhattan plot with genome-wide significant ($p < 5 \times 10^{-08}$) hits indicated for chromosomes 3 and 10. Red line indicates genome-wide significance ($p < 5 \times 10^{-08}$), blue line indicates suggestive significance ($p < 5 \times 10^{-06}$). TNBC triple negative breast cancer, λ Lambda genomic control.

including their potential role as part of risk assessment for stratified breast screening[47].

GWAS from African populations are not only of value not only for the development and understanding of PRS but also to better understand the genetic causes of cancer, which could be of benefit to all. Differences in ancestral origins are associated with differences in allele frequency and linkage disequilibrium patterns. Although this study has a relatively small sample size and is underpowered to detect small effect sizes, there may be risk alleles in populations of African ancestry that are rare or absent in non-African populations and could provide novel insights into our understanding of disease.

Despite the limitation in the sample size of our study we were able to identify two genome-wide significant signals associated with overall BC (rs899342 in the *USP22* gene on chromosome 17 and risk allele rs7181788 on chromosome 15), and several comparably significant signals in our analysis of BC estrogen receptor subtypes. The genomic locations of these signals are interesting in the context of their potential functional significance in the biology of BC, but verification of their relevance will require further bioinformatic and experimental analysis. It is however noteworthy that several of these loci include genes involved in the epithelial-mesenchymal transition, given the important role of that pathway in breast tumor cell progression, invasion, and metastasis. Going forward, the large global Confluence project on the genetics of breast cancer (https://dceg.cancer.gov/research/cancer-types/breast-cancer/confluence-project), to which we are contributing, includes a major expansion in the study of breast cancer genetics in resident African populations.

## Methods
### Study design
This genetic association study forms part of a larger study: Evolving Risk Factors for Cancer in African populations (ERICA-SA) (https://www.samrc.ac.za/intramural-research-units/evolving-risk-factors-cancers-african-populations-erica-sa). Our study received approval from the Human Research Ethics Committee (Medical), University of the Witwatersrand, South Africa for the breast cancer (M160807) and AWI-Gen (M121029; M170880) studies. All the participants signed an Informed Consent Form before any study procedure was performed.

### Study sample
Black female patients with histologically confirmed breast cancer were recruited to the Johannesburg Cancer Study (JCS)[29]. All study participants were enrolled from the Soweto region, Gauteng Province, South Africa. Non-cancer, ethnically similar female participants also from the Soweto region, Gauteng Province were selected from the Africa Wits INDEPTH partnership for genomic studies (AWI-Gen) study and the JCS as population controls[27].

### Sampling and genotyping
We collected and isolated genomic DNA (gDNA) as previously described from peripheral blood samples from all study participants[48]. In brief, gDNA was isolated using either by the Qiagen DNA FlexiGene kit as per the manufacturer's protocol (Cat. No./ID: 51206), or the salting out method in which cellular proteins are salted out by dehydration and precipitation with a saturated NaCl solution[49]. The isolated gDNA was resuspended in low Tris-EDTA buffer and stored at −80 °C until use[50].

DNA samples were genotyped using the Illumina H3Africa custom array (https://www.h3abionet.org/h3africa-chip)[51]. The genotyping of JCS samples took place at the Genomics Core Facility, Department of Social, Genetic & Development Psychiatry Centre, King's College London. The AWI-Gen samples were genotyped using the Illumina FastTrack Sequencing Service (https://www.illumina.com/services/sequencing-services.html). Raw intensity files (iDATs) were used for data analysis. Illumina supplied the predefined cluster file and manifest file which was used to call and cluster the genotypes for all the cases and controls (Supplementary dataset 1). (https://emea.support.illumina.com/downloads/iaap-genotyping-orchestrated-workflow.html#:-:text=Support%20Center%3A,GTC%20format%20and%20PED%20Files). The Illumina Array Analysis Platform Genotyping orchestrated command-line workflow, using the Illumina GenCall algorithm, was used for genotype calling. PLINK version 1.9 was used for genotype data management[52]. The H3ABioNet/H3Agwas Pipeline version 3 was used to format data and carry out data quality control (QC)[53].

Quality control: Only autosomal SNPs were retained for analysis. SNPs were included if SNP-based missingness was ≤0.01, minor allele frequency (MAF) ≥0.01 and Hardy Weinberg equilibrium (HWE) p-value ≥0.0005. Samples with individual genotype missingness ≥0.01 were excluded. Unrelated participants were retained for analysis (piHat ≤0.18). Genotype-gender mismatched individuals were excluded along with participants outside of the heterozygosity limits of ≤0.15 and ≥0.343.

### Imputation
We used the Sanger Imputation Service (https://imputation.sanger.ac.uk/) with the African Genome Resource panel as the reference. Pre-phasing was performed using EAGLE2. Parameters for post-imputation QC were: MAF ≥ 0.01, Impute2 Score ≥ 0.3, HWE p-value ≥ 0.0001.

### Adjusting for population sub-structure
The South African Black populations show complex genetic architecture and population substructure[25,27]. Several measures were taken to account for this. First, admixture analysis was done with reference population of the European, (CEU, $n = 503$), East Asian (EAS, $n = 504$) and South East Asian (SAS, $n = 489$)

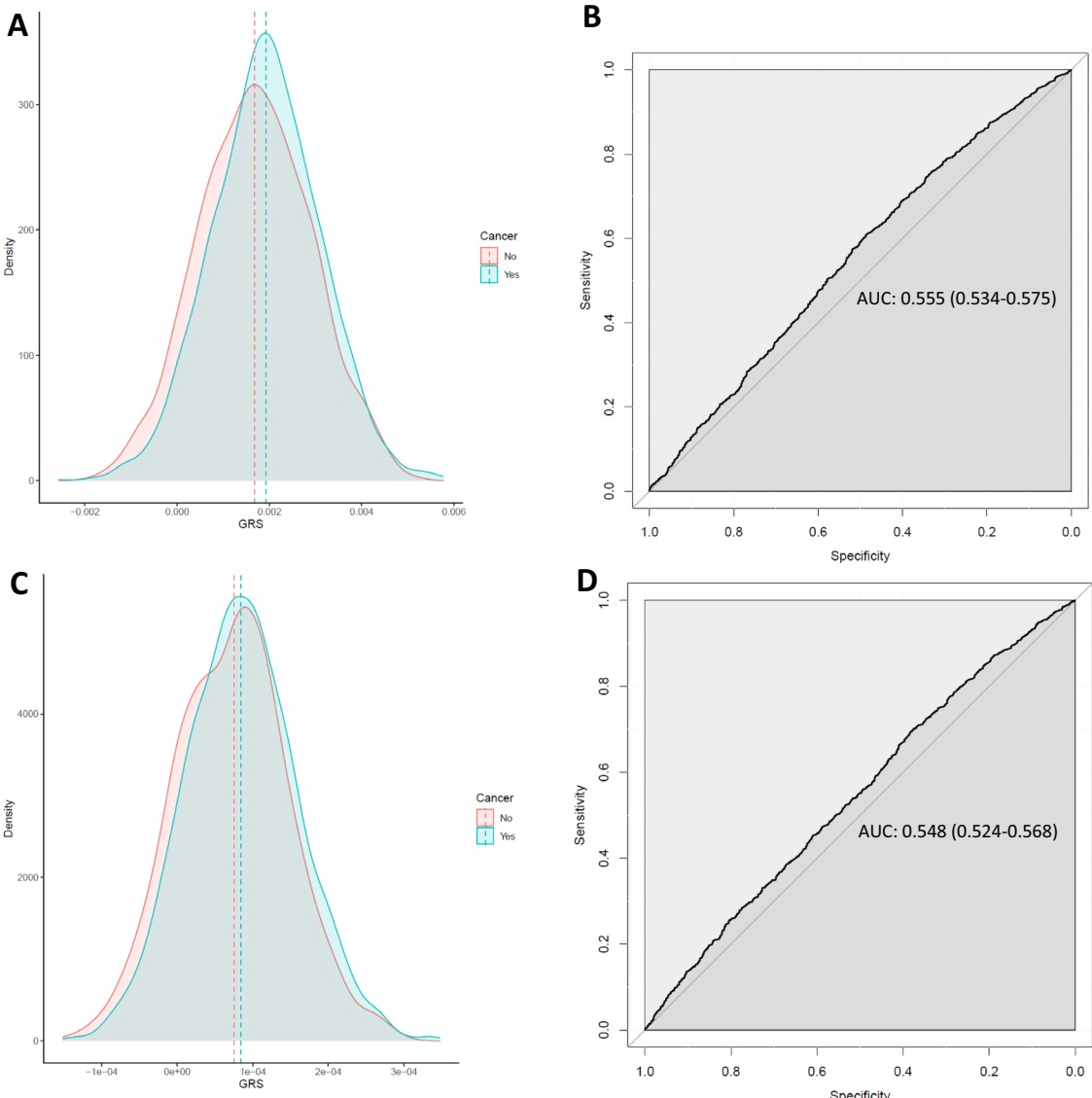

**Fig. 5 | Density and receiver operating characteristic (ROC) plots of non-African polygenic risk scores (PRS) applied to the South African JCS dataset. A** Density plot of score separated by disease status using 202 SNPs PRS. **B** ROC curve plot for 202 SNPs PRS. **C** Density plot of score separated by disease status using 2819 SNPs PRS. **D** ROC curve plot for 2819 SNPs PRS. JCS Johannesburg Cancer Study.

individuals from the 1000 Genomes Project (KGP), 220 individuals with Khoe-San ancestry and 13,261 individuals with SSA ancestry (West, East and South SSA ancestry) from the African Genome Variation Project (AGVP), AWI-Gen and the JCS cohort using Admixture v1.3 51 (see Supplementary Dataset 1a, b)[54–56]. Individuals with >10% CEU or Asian genetic contribution and <70% Bantu and Khoe-San southern sub-Saharan ancestry were excluded. Secondly, we performed Eigen decomposition for Principal Component (PC) analysis using linkage-disequilibrium (LD) pruned SNPs (100 kb window, 20 SNPs within each window, $r^2 = 0.2$). PCs 1–5 were selected using cases and controls after quality control and included as covariates in the final model. Eigen decomposition was performed using PLINK v.1.9 and visualized in R[52].

### GWAS Linear-mixed modeling (LMM)
The binary case-control phenotype was regressed with PCs 1–5 with GRMs and probability of imputation as covariates. The LMM accounts for genetic relatedness and population structure and was done using Gemma v.0.98.1[57,58] GRMs were generated using 500,000 LD

independent genotyped SNPs using the leave-one-chromosome-out (LOCO) approach. Study methodologies incorporating mixed models that utilize the LOCO approach have higher statistical power compared to traditional association studies[59,60]. Odds ratio approximations were calculated using case-control ratios and beta values[61]. The quantile-quantile (QQ) plots and Manhattan plots were done using the fastman library in R[26,62].

### Receptor sub-type analysis
Receptor sub-type analysis was done with ER positive cases and ER negative cases against controls, TNBC cases against: controls; ER-positive cases and HER2-positive cases. An analysis was also done comparing ER-positive individuals (coded as 1) with ER-negative individuals (coded as 0). Sample sizes for the receptor subtypes are shown in Table 1.

We also assessed replication of suggestive signals ($p < 5 \times 10^{-06}$) from the JCS ER-negative vs controls results in the Jia et al. ER-negative vs controls dataset[17]. Further signals ($p < 5 \times 10^{-06}$) from the ER-negative and ER-positive vs controls analysis from Jia et al. were looked up in our JCS ER-negative and ER-positive dataset[17].

## Heritability estimation

A SNP-based heritability ($h2g$) estimate was calculated in LDAK using genotype data[63]. A restricted maximum likelihood estimations (REMLs) was used in LDAK. LDAK weighting, which accounted for LD, was carried out using the default correlation squared threshold of 0.98. A GRM was computed on the smaller set of predictors that resulted from the LDAK weighting, and this was used for the $h2g$ estimation. The $h2g$ for BC was estimated on the liability scale using the Globocan 2020 incidence for BC (age standardized incident rate of 0.000526) in South Africa as a proxy for disease prevalence[1].

## Replication of JCS African and known BC risk loci

In order to determine whether our findings could be replicated in existing BC GWAS data from other populations we first performed a fixed-effect meta-analysis in METAL, allowing for heterogeneity, on two datasets: African ancestry BC cases and ethnically-matched controls from the UK Biobank (cases = 163, controls = 3774), and the dataset from the Jia et al. (2024) GWAS (cases = 18,034, controls = 22,104) (Table 1)[17,64,65]. Suggestive signals in our study ($p < 5 \times 10^{-06}$) were then assessed for replication in this African ancestry meta-analysis[17,64].

Suggestive signals ($p < 5 \times 10^{-06}$) from the African BC GWAS by Jia et al. were tested for replication in the JCS BC dataset.

## Fine mapping & functional analysis of associated variants

Regional plots were created using LocusZoom v1.4, for all top GWAS signals with $p < 5 \times 10^{-8}$, with a 400 kb flanking nucleotide window, using KGP African LD information[66]. FUMA was used to annotate[67,68] and interpret associated GWAS variants with $p < 1 \times 10^{-5}$ using the KGP Phase 3 African data as a reference, as well as annotated co-localized eQTLs in the breast tissues of interest from GTEx version 8[69,70]. GCTA COJO-SLCT was used to perform a stepwise model selection procedure to select independently associated SNPs and FINEMAP v1.4 was used to identify variants surrounding the top association signals in our study and credible interval set at 95%. The top SNPs were also analyzed on PancanQTLv2.0, which provides cis and trans eQTLs in 33 cancer types from The Cancer Genome Atlas[71]. Reactome was used to investigate pathways linked to the genes that were located near the two top signals[72].

## Polygenic risk scores

A PRS was generated using PLINK in our dataset using the 313 SNP model by Mavaddat et al.[33]. Of the 313 SNPs, 202 were present in our dataset and used to generate the PRS. In addition, we also generated a PRS using the 3820 SNPs model by Mavaddat et al.[33]. Of the 3820 SNPs, 2819 SNPs were present in our dataset and were used to generate the PRS. PRS and cancers status were compared using logistic regressions (*lm* function from R) including PCs 1–5 as covariates. The percentage of variance explained by the PRS of cancer status was estimated using the linear model (*lm*) from R and the ANOVA function. Only SNPs with an allele frequency of >0.01 were included in this analysis. The discrimination performance of a PRS was assessed using the area under the receiver operating characteristic curve (AUC), using roc function from pROC package in R[73].

## Reporting summary

Further information on research design is available in the Nature Portfolio Reporting Summary linked to this article.

## Data availability

The full dataset generated in this study is in the EGA database under the study accession code EGAS00001002482 for AWI-Gen controls and EGAS00001008032 for breast cancer cases and JCS controls. This accession IDs for the AWI-Gen phenotype dataset: EGAD00001006425, and the genotype dataset: EGAD00010001996. These datasets are available subject to controlled access through the Data and Biospecimen Access Committee of the H3Africa Consortium. Summary statistics reported in the paper are accessible on GWAS Catalog (https://www.ebi.ac.uk/gwas/) at the accession numbers: GCST90551892, GCST90551893, GCST90551894, GCST90551895, GCST90551896, GCST90551897, GCST90551898. Publicly available datasets included in the study are the following: 1000 Genomes Project Phase 3 (ftp://ftp.1000genomes.ebi.ac.uk/vol1/ftp), BC African American dataset with summary statistics available at GWAS Catalog (https://www.ebi.ac.uk/gwas/). The data will be available for computational benchmarking studies on condition that no attempt is made to reidentify participants. Access to the dataset will require ethics approval from a recognized ethics committee.

## Code availability

The quality control pipeline is available on GitHub at https://github.com/h3abionet/h3agwas/. The version used in this study has been deposited in the Zenodo repository (https://doi.org/10.5281/zenodo.14907702; https://zenodo.org/records/14907702). Additional code is available upon request.

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

## Acknowledgements

We acknowledge the contributions of Prof Ananyo Choudhury and Dr Dhriti Sengupta, Prof Scott Hazelhurst for access to the Wits Computer cluster and AWI-Gen data, the SBIMB biobank and personnel for handling of our samples, the contributions of our field workers, phlebotomists, laboratory scientists, administrators, data personnel and all other staff who contributed to the data and sample collections, processing and storage. We also acknowledge the NIH-funded H3Africa Consortium Collaborative Centre, AWI-Gen, for sharing data for the population controls used in our study (PI: M.R.). It is with great sorrow that we record the death of our colleague and co-author Dr. Elvira Singh on 27 February 2022. We acknowledge Sisters Gloria Mokwatle, Patricia Rapoho and Pheladi Kale who carried out the interviews and collected blood specimens from patients. We thank the oncology clinicians and administration at Charlotte Maxeke Johannesburg Academic Hospital for assistance and access to patients in their care, and the patients who gave freely of their time. The ERICA-SA Study and the JCS were supported by the South African Medical Research Council with funds received from the South African National Department of Health and the UKMRC (with funds from the UK Government's Newton Fund) (MRCRFA-SHIP 01-2015). Additional support was received from the Cancer Association of South Africa (CANSA2022 to C.G.M), the National Health Laboratory Service (to E.S.) and the NIHR Maudsley Biomedical Research Centre, Maudsley NHS Foundation Trust and King's College London (NIHR203318 to C.M.L.). The contents of this publication are solely the responsibility of the authors and do not necessarily represent the official views of the SAMRC or the South African National department of health.

## Author contributions

M.H., W.C.C., C.B.d.V., C.M.L., D.B., F.S., E.S., C.G.M. and J.T.B. designed the study. C.G.M., D.B., F.S., E.S., T.W., R.N., C.B.d.V., C.M.L. and M.R. acquired the funding. W.C.C., C.B.d.V., S.L., C.C., F.S., M.R., M.M. and E.S. were responsible for sample acquisition, processing, and management and genotyping. M.H., W.C.C. and J.T.B. performed the analysis. M.H., W.C.C., C.G.M. and J.T.B. wrote the manuscript. All authors critically reviewed, edited, and approved the manuscript.

## Competing interests

The authors declare no competing interests.
