## [Transparent Peer Review file · Nature Communications]

Genome-wide association study identifies common variants associated with breast cancer in South African Black women

Corresponding Author: Dr Mahtaab Hayat

Version 0:

Reviewer comments:

Reviewer #1

(Remarks to the Author)

The GWAS study has been widely used to discover risk SNPs for human diseases such as cancer, primarily focusing on European populations. However, information on African populations remains severely limited. In this study, the authors conducted a GWAS on a cohort of 2,485 African breast cancer patients and 1,101 controls, identifying two previously unreported risk loci. One locus, situated between the RAB27A and UNC13C genes (rs7181788, $p = 1.01 \times 10^{-8}$), is associated with oncogenesis and tumor progression.

The reviewers commend this research initiative, believing that insights from African populations will offer novel genetic perspectives on breast cancer. However, a major limitation is the relatively small sample size (2,485 cases vs. 1,101 controls). The authors acknowledged this challenge in their manuscript. Consequently, the identified SNPs could not be replicated in the African-American ancestry cohort from BCAC, reflecting similar challenges in replication studies. Well, the lack of shared risk loci might be attributed to genetic differences between African populations, particularly West African ancestry predominant in African Americans versus the South African ancestry in this study. Despite the intriguing findings, the current study's sample size remains insufficient to draw definitive conclusions. Even though, for an initial discovery of common variants with moderate effect sizes, a sample size of 2000 cases and 1000 controls may be sufficient for publication.

The manuscript would benefit from including PCA plots displaying all reference populations, especially West African (KGP and AWI-Gen) and African American (KGP), alongside the study population (Figure 1a). Given the heterogeneous nature of African populations comparing to other major geographically structured populations, understanding these differences is crucial, particularly in light of the challenges in SNP replication observed in the BCAC study.

For subtype analysis, the authors may consider stratifying TNBC vs. ER vs. HER2 status, although potential limitations due to sample size. This stratification is important, as clinical outcomes in TNBC among African Americans are significantly worse compared to European Americans.

Reviewer #2

(Remarks to the Author)

This is an important paper that has the potential to contribute to our understanding of genetic predisposition to breast cancer in African population which is genetically diverse and often understudied. However, the manuscript can benefit from clarifying the following points:

1. Line 79-80: For clarify, authors are advised to include brief description on the difference between AA and SSA to justify the impact of the research. Is AA genetically distinct from the AA?
2. Line 83: for comparison, can you please also include the range of sample size of existing AA studies?
3. Line 114: Why use LMM? Is it because of the imbalance dataset?
4. Line 115: Can you also generate PC plot for AA and SAA?
5. Line 119: do you mean genotyping rate?
6. Line 131: "Both signals are supported by other SNPs in these regions". Do you mean there are multiple correlated SNPs

in the region? Please rephrase.

7. Line 138: Please include the regional association plot for rs9516904

8. Line 139: Please provide the allele frequency in European, Asian and African population.

9. Line 150: Why were the other SNPs (91-37) not available? Is it because of poor imputation score in BCAC AA, or low allele frequency because of the genetic differences in AA and SAA?

10. Line 151-154: Were these three SNPs (rs7181788, rs899342, rs9516904) replicated or among the 37? If they are not among the 37, it is worth to explore further why were they not in BCAC AA.

11. Line 162: What is the sample size of AA in BCAC and also UK Biobank? Why not consider UK Biobank as replication cohort like BCAC AA?

12. Line 165: Please include the OR from the original GWAS, the results seems to be driven by the original GWAS. Actually I am not sure what's the value of doing meta-analysis. Why not combine BCAC AA and UKBiobank as larger replication cohort.

13. Line 173: What's the sample size of ER+ and ER- in this cohort? Table 3 shows different SNPs identified through the three GWASs. Usually one would either do the case control analyses, or ER-specific vs controls analyses. What can we conclude from Table 3?

14. Line 228: need to give better clarification on why it is important to run GWAS of resident African populations.

15. Line 240-242: The results was driven by the original GWAS.

16. What's the association of the chr15 and 17 SNPs in European population? Were they not picked up previously because of low frequency in European/Asian?

17. For subtype analysis, can you evaluate the candidate SNPs in BCAC AA?

18. Discussion: too lengthy.

Version 1:

Reviewer comments:

Reviewer #1

(Remarks to the Author)

My questions have been fully addressed.

Reviewer #2

(Remarks to the Author)

The authors have satisfactorily addressed all of my comments, and I have no further feedback.

Hayat et al: Response to Reviewers

We thank the reviewers for their helpful and constructive comments. A detailed response to the points raised is given in the table below. We would also like to explain the significant changes we have made to the replication section of the manuscript.

While our paper was under review a large study of African breast cancer was published by Jia et al in Nature Genetics (56: 819-826). This included a GWAS of 18,034 cases and 20,104 controls of African ancestry, with 85.3% participants being African-Americans and the remainder from Barbados and the African continent (Nigeria and Ghana). We therefore decided to combine this large dataset, which included African data from the BCAC consortium, with African breast cancer data from the UK Biobank in a meta-analysis which represented all known GWAS data on African breast cancer. We then used the results of this meta-analysis, which we refer to as the African Ancestry (AA) dataset, to look for replication of the association signals in our South African (SA) study, and to look for replication of the AA signals in our South African study. As can be seen in the revised manuscript there was little evidence of replication of our signals in the AA data but some replication of AA signals in our SA data. We note that essentially all the large existing datasets on African breast cancer are of West African origin. Given the substantial genetic differences between populations of West and South Africa which are documented in our paper, it is not unreasonable to suppose that there may be significant differences in the genetic variants which are associated with breast cancer risk in these populations.

Reviewer 1	
Comment	Response
The manuscript would benefit from including PCA plots displaying all reference populations , especially West African (KGP and AWI-Gen) and African American (KGP), alongside the study population (Figure 1a). Given the heterogeneous nature of African populations comparing to other major geographically structured populations, understanding these differences is crucial, particularly in light of the challenges in SNP replication observed in the BCAC study.	Added PCA plot with reference populations – Figure 1B. Lines 131-133 state that “The PC plot showed that the South African BC cases and controls were well matched and clustered away from non-South African samples and that West African populations are distinct from South African populations.”
For subtype analysis, the authors may consider stratifying TNBC vs. ER vs. HER2 status, although potential limitations due to sample size. This stratification is important, as clinical outcomes in TNBC among African Americans are significantly worse compared to European Americans.	Triple negative breast cancer (TNBC) cases (n=262) were analyzed against controls (n=1,101); ER+ (n=1,143) and HER2+ (n=499). Sample sizes are given in Table 1. Significant results and replication tests have been included in the text (lines 192–232), in Table 3 and in supplementary tables S7 and S8.
Reviewer 2	
Comment	Response
1. Line 79-80: For clarify, authors are advised to include brief description on the difference between AA and SSA to justify	A statement has been added statement to the Introduction, lines 102-109: “Genotyping and whole genome

the impact of the research. Is AA genetically distinct from the AA?	sequencing studies have revealed a very high degree of genetic diversity among the populations of the African continent, with principal component analysis showing clear separation of populations from West, East, Central and Southern Africa. ²⁵ African-Americans originated from Africans forced into slavery and are descended mostly from ethnic groups that lived in West Africa, with admixture mostly of European ancestry. ²⁶ It is therefore likely that genetic studies of breast cancer in African-Americans will capture only a subset of the contribution of the genetic contribution to breast cancer susceptibility on the African continent, and argues for broadening the diversity of genetic studies in Africa.” These differences are also shown in our own PCA and admixture data in Figure 1 and described in the Results (lines 129-133).
2. Line 83: for comparison, can you please also include the range of sample size of existing AA studies?	Responded: Sample size ranges are now included in lines 84-86 and 90-93.
3. Line 114: Why use LMM? Is it because of the imbalance dataset?	Added reason, lines 154-155: “LMM method was used because it is effective in correcting for relatedness and structure, therefore limiting genetic inflation.”
4. Line 115: Can you also generate PC plot for AA and SAA?	As requested by Reviewer 1 we added this as figure 1B
5. Line 119: do you mean genotyping rate?	Corrected to genotyping rate – line 149.
6. Line 131: “Both signals are supported by other SNPs in these regions”. Do you mean there are multiple correlated SNPs in the region? Please rephrase.	Yes. We have corrected this to “there are multiple correlated SNPs in the region of both signals” (line 161).
7. Line 138: Please include the regional association plot for rs9516904	On reflection from this comment we have removed this sentence as the regional association plot does not show strong support from other SNPs in LD with it – line 169.
8. Line 139: Please provide the allele frequency in European, Asian and African population.	Since the sentence regarding this SNP has been removed (see point 7 above) we have removed the comment regarding allele frequencies as it is no longer relevant.
9. Line 150: Why were the other SNPs (91-37) not available? Is it because of poor imputation score in BCAC AA, or low allele frequency because of the genetic differences in AA and SAA?	The assessment of replication of JCS BC GWAS signals in the AA data from BCAC has been removed and replaced by assessment of replication in a meta-analysis of the new African data from Jia et al (2024) and African data from UK Biobank (lines 177-183). See also introductory comments above.

10. Line 151-154: Were these three SNPs (rs7181788, rs899342, rs9516904) replicated or among the 37? If they are not among the 37, it is worth to explore further why were they not in BCAC AA.	These SNPs were not replicated in the new meta-analysis of data from Jia 2024 and UK Biobank (lines 177-183).
11. Line 162: What is the sample size of AA in BCAC and also UK Biobank? Why not consider UK Biobank as replication cohort like BCAC AA?	The new meta-analysis includes African data from BCAC and UK Biobank and has now been used as a replication cohort. The sample sizes are now given in Table 1.
12. Line 165: Please include the OR from the original GWAS, the results seems to be driven by the original GWAS. Actually I am not sure what's the value of doing meta-analysis. Why not combine BCAC AA and UK Biobank as larger replication cohort.	The meta-analysis of the South African JCS data with the Africa data from BCAC and from the UK Biobank has been removed. As mentioned in the reply to point 9 above we have done a meta-analysis with the new data from Jia et al 2024 (which includes the Africa BCAC data) and the data from UK Biobank, and used this as a large replication cohort (lines 176-191).
13. Line 173: What's the sample size of ER+ and ER- in this cohort? Table 3 shows different SNPs identified through the three GWASs. Usually one would either do the case control analyses, or ER-specific vs controls analyses.	A new Table 1 has been added with sample sizes. We have done analyses of receptor subtypes vs controls and also an ER+ vs ER- analysis to look for potential differences between these subtypes.
14. Line 228: need to give better clarification on why it is important to run GWAS of resident African populations.	Added: Lines 262-265: "The immense genetic diversity among the populations of sub-Saharan Africa and differences in environmental exposures between resident and non-resident African populations suggests that there may be substantial differences in the genetic determinants of cancer susceptibility both within continental Africa and across continents." See also our comments on potential reasons for limited replication, lines 318-324.
15. Line 240-242: The results was driven by the original GWAS.	This analysis and associated comments have been removed as per point 9 above.
16. What's the association of the chr15 and 17 SNPs in European population? Were they not picked up previously because of low frequency in European/Asian?	No associations at these loci have been found in European populations. Allele frequencies are fairly similar in European and Asian populations.
17. For subtype analysis, can you evaluate the candidate SNPs in BCAC AA?	The top SNPs from our subtype analysis were evaluated in the Jia et al (2024) subtype data set. The JCS signals for TNBC and ER- were not replicated in the Jia data, but there was some evidence for replication of Jia et al signals in our SA JCS data (lines 224-233).

18. Discussion: too lengthy.

Amended and shortened to remove some repetitive comments.